# Genetic risk score for insulin resistance based on gene variants associated to amino acid metabolism in young adults

Eunice Lares-Villaseñor[1☯], Martha Guevara-Cruz[2☯], Samuel Salazar-García[1], Omar Granados-Portillo[2], Mariela Vega-Cárdenas[3], Miguel Ernesto Martinez-Leija[1], Isabel Medina-Vera[4], Luis E. González-Salazar[5], Liliana Arteaga-Sanchez[2], Rocío Guízar-Heredia[2], Karla G. Hernández-Gómez[2], Aurora E. Serralde-Zúñiga[5], Edgar Pichardo-Ontiveros[2], Adriana M. López-Barradas[2], Laura Guevara-Pedraza[6], Guillermo Ordaz-Nava[2], Azalia Avila-Nava[7], Armando R. Tovar[2], Patricia E. Cossío-Torres[8], Ulises de la Cruz-Mosso[9], Celia Aradillas-García[10], Diana P. Portales-Pérez[1], Lilia G. Noriega[2]*, Juan M. Vargas-Morales[1]*

**1** Facultad de Ciencias Químicas, Universidad Autónoma de San Luis Potosí, San Luis Potosí, México, **2** Fisiología de la Nutrición, Instituto Nacional de Ciencias Médicas y Nutrición Salvador Zubirán, Ciudad de México, México, **3** Laboratorio de Nutrición, Departamento de Ciencias en Investigación Aplicadas en Ambiente y Salud, Coordinación para la Innovación y Aplicación de la Ciencia y la Tecnología, Universidad Autónoma de San Luis Potosí, San Luis Potosí, México, **4** Departamento de Metodología de la Investigación, Instituto Nacional de Pediatría, Ciudad de México, México, **5** Servicio de Nutriología Clínica, Instituto Nacional de Ciencias Médicas y Nutrición Salvador Zubirán, Ciudad de México, México, **6** Universidad Anáhuac México, Ciudad de México, México, **7** Hospital Regional de Alta Especialidad de la Península de Yucatán, IMSS-Bienestar, Mérida, Yucatán, Mexico, **8** Departamento de Salud Pública y Ciencias Médicas, Facultad de Medicina, Universidad Autónoma de San Luis Potosí, San Luis Potosí, México, **9** Red de Inmunonutrición y Genómica Nutricional en las Enfermedades Autoinmunes, Centro Universitario de Ciencias de la Salud, Universidad de Guadalajara, Guadalajara, México, **10** Facultad de Medicina, Coordinación para la Innovación y Aplicación de la Ciencia y la Tecnología, Universidad Autónoma de San Luis Potosí, San Luis Potosí, México

☯ These authors contributed equally to this work.
* lilia.noriega@incmnsz.mx (LGN); juan.vargas@uaslp.mx (JMVM)

## Abstract

Circulating concentration of arginine, alanine, aspartate, isoleucine, leucine, phenylalanine, proline, tyrosine, taurine and valine are increased in subjects with insulin resistance, which could in part be attributed to the presence of single nucleotide polymorphisms (SNPs) within genes associated with amino acid metabolism. Thus, the aim of this work was to develop a Genetic Risk Score (GRS) for insulin resistance in young adults based on SNPs present in genes related to amino acid metabolism. We performed a cross-sectional study that included 452 subjects over 18 years of age. Anthropometric, clinical, and biochemical parameters were assessed including measurement of serum amino acids by high performance liquid chromatography. Eighteen SNPs were genotyped by allelic discrimination. Of these, ten were found to be in Hardy-Weinberg equilibrium, and only four were used to construct the GRS through multiple linear regression modeling. The GRS was calculated using the number of risk alleles of the SNPs in *HGD*, *PRODH*, *DLD* and *SLC7A9* genes. Subjects with high GRS ($\geq$ 0.836) had higher levels of glucose, insulin, homeostatic model assessment- insulin resistance (HOMA-IR), total cholesterol and triglycerides, and lower levels of arginine than subjects with low GRS ($p < 0.05$). The application of a GRS based on variants

**Data Availability Statement:** The data underlying the results presented in the study are available at https://doi.org/10.6084/m9.figshare.24968322.

**Funding:** This work was supported by CONACYT-202721 and CONACYT-CF2019-2096049 to LGN, by CONACYT-PN-2016-01-3324 to MGC, and by consultancy and industrial services 17 and 34 to JMVM from Facultad de Ciencias Químicas from the Universidad Autónoma de San Luis Potosí. There was no additional external funding received for this study. The funders had no role in study design, data collection and analysis, decision to publish, or preparation of the manuscript.

**Competing interests:** The authors have declared that no competing interests exist.

within genes associated to amino acid metabolism may be useful for the early identification of subjects at increased risk of insulin resistance.

## Introduction

The presence of different cardiometabolic risk factors and components of the metabolic syndrome such as obesity, dyslipidemia, hypertension, hyperglycemia and insulin resistance (IR) increase the risk of cardiovascular disease and type 2 diabetes (T2D) [1, 2]. Notably, changes in the circulating amino acid profile are related to IR [3]. In fact, subjects with IR have higher serum concentrations of branched chain amino acids (BCAA), aromatic amino acids (AAA), glutamine, glutamate and lower levels of glycine than subjects without IR [4]. Even young adults with IR have higher plasma concentration of arginine, alanine, aspartate, isoleucine, leucine, phenylalanine, proline, tyrosine, taurine and valine [5]. Moreover, different epidemiological studies have reported that BCAA (isoleucine, leucine and valine), AAA (phenylalanine and tyrosine) and glutamine could predict the development of T2D [6–8].

Amino acid levels change with age and the presence of SNPs involved in BCAA metabolism in subjects with obesity and MetS [9, 10]. Interestingly, the joint presence of two SNPs, the *BCAT2* (Branched Chain Amino Acid Transaminase 2) rs11548193 and *BCKDH* (Branched Chain Keto Acid Dehydrogenase) rs45500792, has higher circulating levels of aspartate, isoleucine, methionine, and proline than the subjects homozygotes for the most common allele [11]. This evidence reflects that the sum of various risk alleles may provide a better estimation of plasma amino acid levels and IR. Actually, genome-wide association studies (GWAS) have identified multiple SNPs that influence serum concentrations of circulating metabolites, including amino acids such as BCAA, AAA, histidine and glutamine [12]. This has led us to speculate whether the presence of different SNPs related to amino acid metabolism could alter their plasma concentrations and, moreover, help us to predict subjects with higher risk to develop IR. This could be achieved through the development of a GRS, which represents the cumulative contribution of risk alleles from various SNPs on a specific outcome of interest within an individual. Combining several variants into a GRS can capture an individual's susceptibility to a disease [13–15]. Therefore, the aim of this study was to develop a GRS to predict the risk of IR in young Mexican adults based on the determination of some selected SNPs of genes related to the metabolism of amino acids.

## Methods

### Study population

We carried out a cross-sectional study. Subjects from the general population who were carrying out university admission procedures were invited to participate in the study. Subjects were recruited from June 16th, 2014 to July 3rd, 2014, at the Universidad Autónoma de San Luis Potosí (UASLP) in San Luis Potosí, México. Subjects received detailed information about the study, and those who wished to participate gave their written informed consent. The identity information of all patients was coded to ensure that privacy was not compromised. The study was designed in accordance with the Declaration of Helsinki and the ethical treatment of human subjects, and was approved by the Ethics Committee of the National Institute of Medical Sciences and Nutrition Salvador Zubirán (Registration number 669). The participants were Mexicans from 18 to 25 years old with body mass index (BMI) $\geq$ 18.5 and $<$ 40 kg/m$^2$. The exclusion criteria included pregnancy, substance abuse, history of cardiovascular events,

chronic diseases (including individuals previously and newly diagnosed with T2D), and treatment with hypoglycemic, antihypertensive agents, agents used to treat dyslipidemias, steroids, and immunosuppressors. The elimination criterion was the voluntary withdrawal of the participants. Subjects were evaluated by medical examination to collect anthropometric and clinical measurements, and underwent blood sample collection for biochemical analysis, DNA extraction and SNPs determination. Among the biochemical variables, amino acids were determined (alanine, arginine, aspartic and glutamic acids, glycine, histidine, isoleucine, leucine, lysine, methionine, phenylalanine, proline, serine, threonine, tyrosine and valine).

## Anthropometric measurements

Anthropometric evaluation was performed after a 12-h fast. Height was measured with a mobile SECA [R] stadiometer (Seca 213, USA), and weight was obtained twice using a TANITA [R] calibrated electronic device (Tanita UM-081, Kyoto, Japan). Body mass index (BMI) was calculated using Quetelet's formula:

$$BMI = \frac{Weigh\ (kg)}{Height\ (m)^2} \tag{1}$$

The subjects were classified according to the BMI based on the World Health Organization [16].

## Clinical measurements

Systolic blood pressure (SBP) and diastolic blood pressure (DBP) measurement was performed in the dominant right arm and in a sitting position using an OMRON [R] digital sphygmomanometer (HEM-7130, Kyoto, Japan) and appropriately sized cuffs according to clinical standards [17]. We considered altered blood pressure with a cut-off point of $\geq 130/85$ [18].

## Biochemical analyses

Blood samples were collected from the subjects after a 12-hour fast, and serum was subsequently extracted. Glucose, total cholesterol, high density lipoprotein cholesterol (HDL-C), low density lipoprotein cholesterol (LDL-C) and triglycerides were measured enzymatically using an Ortho Clinical Vitros 250 Chemistry System ([C]Ortho-Clinical Diagnostics, Inc. Raritan, NJ.). Insulin and leptin were measured by radioimmunoassay (RIA, Millipore, Billerica, MA, USA). IR was obtained through the HOMA-IR [19].

$$HOMA - IR = \frac{Glucose\ \left(\frac{mg}{dL}\right) * Insulin\ \left(\frac{\mu U}{mL}\right)}{405} \tag{2}$$

IR was established with a HOMA-IR value $\geq 2.5$ [20, 21]. Amino acids were measured by high performance liquid chromatography (HPLC). Briefly, 50 μL of 10% sulfosalicylic acid was added to 200 μL of serum, incubated for 30 min at 4 ˚C, and centrifuged at 14,000 rpm for 20 min. The supernatant was obtained and one microliter of internal standard (25 mM norvaline) was added prior to derivatization and injection. For derivatization, o-phthalaldehyde (OPA) and 9-fluorenylmethyl chloroformate (FMOC) were used. Derivatization and injection were carried out using a sampling device (Agilent; G1367F) coupled to an Agilent 1260 Infinity HPLC with fluorescent detector (Agilent; G1321B). A ZORBAX Eclipse AAA column was used at 40 ˚C and the chromatographic conditions indicated by the manufacturer (Agilent; 5980–1193) were applied [22].

## SNP selection

We performed a bibliographic search to identify SNPs present in genes related to amino acid metabolism (such as catabolic enzymes or amino acid transporters), which were previously associated with alterations in plasma amino acids concentration and/or with cardiometabolic risk factors. The SNPs were selected when a frequency > 10% was reported for the Mexican or Latino population using data managers such as: Genecards (https://www.genecards.org/), GWAS catalog (https://www.ebi.ac.uk/gwas/), DisgeNet (https://www.disgenet.org/) and NCBI dbSNP (https://www.ncbi.nlm.nih.gov/snp/). Additionally, non-synonymous SNPs were preferentially selected. Finally, 18 SNPs that met the selection criteria were chosen (Table 1 in S1 Appendix).

## Genotyping

The buffy coat was extracted from 5 mL of whole blood by centrifugation following established procedures [23]. DNA isolation was performed with the mini kit QIAamp® DNA Blood Mini (QIAGEN, Hilden, Germany). The DNA was adjusted to a concentration of 10 ng/μL. The SNPs assessed were as follows: *TAT* (Tyrosine Aminotransferase) rs74344827 (C_102105963_10); *HGD* (Homogentisate 1,2-Dioxygenase) rs2255543 (C__22272709_10); *GSTZ1* (Glutathione S-Transferase Zeta 1) rs1046428 (C__25922638_20); *GPT* (Glutamic-Pyruvate Transaminase) rs1063739 (C___1922246_20); *OTC* (Ornithine Transcarbamylase) rs1800321 (C__26644158_20); *ASPG* (Asparaginase) rs1744284 (C___7504721_10); *HAL* (Histidine Ammonia-Lyase) rs7297245 (C__25981584_20); *BCAT2* rs11548193 (C__25473139_20); *BCKDH* rs45500792 (C__25600774_10); *PRODH* (Proline Dehydrogenase 1) rs5747933 (C_175679135_20); *DLD* (Dihydrolipoamide Dehydrogenase) rs6943999 (C___3268599_10); *SHMT1* (Serine Hydroxymethyltransferase 1) rs1979277 (C___3063127_10); *MTR* (5-Methyltetrahydrofolate-Homocysteine Methyltransferase) rs1805087 (C__12005959_10); *SLC1A4* (Solute Carrier Family 1 Member 4) rs759458 (C___2681351_10); *SLC7A9* (Solute Carrier Family 7 Member 9) rs1007160 (C___2885706_1_); *PPM1K* (Protein Phosphatase, $Mg^{2+}/Mn^{2+}$ Dependent 1K) rs9637599 (C___9510031_10) and rs1440581 (C___9509992_10); and *GCKR* (Glucokinase Regulator) rs1260326 (C___2862880_1_).

These 18 SNPs were analyzed using allelic discrimination assays using TaqMan probes (AppliedBiosystems®) by the real time polymerase chain reaction (RT-PCR) on a LightCycler® 480 instrument (Roche®). Briefly, a master mixture was prepared considering for each sample 0.75 μL of TaqMan probe, 0.25 μL of molecular grade nuclease-free ultrapure water (USB®, USA), and 5 μL of Probes Master (LightCycler® 480), following the manufacturer's instructions. Then, to perform PCR, 4 μL of previously adjusted DNA and 6 μL of the master mixture were added to each well of the 96-well plates (Roche®). Negative controls were also included, which only carried the master mixture and nuclease-free water. The reactions were performed in duplicate. The cycling conditions consisted of an initial pre-incubation cycle at 95 ˚C for 10 min, followed by 45 cycles of denaturation at 95 ˚C for 12 s, annealing at 60 ˚C for 50 s and extension at 72 ˚C for 2 s and a cooling cycle at 40 ˚C for 30 s. For allelic discrimination results, the context sequence for each Taqman probe and the fluorophores targeting each allele were previously verified based on information reported by the manufacturer.

## GRS calculation

We constructed a multilocus GRS using the SNPs (n = 10) that were in Hardy-Weinberg equilibrium ($p > 0.05$) (Table 2 in S1 Appendix). The GRS was calculated for each individual as the sum of the number of IR risk alleles based on the highest HOMA-IR value (preference cut-

off $\geq$ 2.5). Thus, we developed a simple GRS in apparently healthy subjects, using the allele of the SNP that, according to our hypothesis and what was reported, would influence the risk of IR [9, 24–26] (Table 3 in S1 Appendix). Briefly, for each of the SNPs, we assigned a value of 0, 1, or 2, with the highest value representing homozygotes with the high-risk alleles for IR, and the lowest value representing homozygotes with the low-risk alleles. A value of 1 denoted heterozygotes. Multiple linear regression was performed to develop a GRS for IR risk, using the HOMA-IR value as the dependent variable and each SNP as an independent variable. This process resulted in distinct models. The model of SNPs exhibiting a significant association was selected. Among these SNPs, three were found to be significantly associated with IR ($p < 0.05$), while one exhibited marginal significance ($p = 0.057$).

Thus, only four SNPs served as predictors of IR risk and were used to calculate a weighted GRS for each individual. This involved multiplying the standardized β coefficient by the effect size (0, 1 or 2) for each SNP, followed by summing the scores obtained from the four SNPs for each subject.

$$GRS = \sum_{i=1}^{K} \beta_i \cdot N_i \tag{3}$$

Where $k$ is the number of independent genetic variants associated with IR, $N_i$ corresponds to the effect size (0, 1 or 2) for each SNP, that is, the number of risk alleles for each individual ($i = 1$), and β is the coefficient estimated for each SNP associated with the HOMA-IR.

## Statistical analyses

Continuous variables were presented as median and interquartile range (25th-75th percentiles) or as a mean and standard deviation. These variables were evaluated using the Kolmogorov-Smirnov Z Test to analyze their distribution. The dichotomous or nominal variables were expressed as frequencies and percentages. The Student T test was used for variables with a parametric distribution, while the Mann-Whitney U test was used for non-parametric variables to analyze differences in anthropometric, clinical, and biochemical data. Genotype frequencies were analyzed using chi-square analysis to assess Hardy-Weinberg equilibrium ($p > 0.05$).

For GRS, the effect of each SNP on the HOMA-IR variable was first assessed using a general linear model adjusted for age, sex and BMI. Then, multiple linear regression analysis was used to assess the association between HOMA-IR (dependent variable) and the 10 SNPs (independent variables). Non-collinearity was previously evaluated between the independent variables. The backward-stepwise method was used to select the final model. Significant SNPs were used for the GRS. Moreover, we evaluated the association between the obtained GRS and the HOMA-IR variable adjusting for age, sex, and BMI using a generalized linear model. Subsequently, the GRS was categorized into tertiles. This categorization was used to assess the trends in each anthropometric, clinical, and biochemical variable among the subjects using the Jonckheere-Terpstra test.

Lastly, ANOVA and Bonferroni post-hoc test with and without adjustment for covariates (age, BMI and sex) were used to assess differences in the variables of interest and the GRS. Previously, the nonparametric data were logarithmically transformed. Differences were considered significant at $p < 0.05$. Data were analyzed using SPSS software version 20.0 (SPSS Inc., USA).

## Results

### Characteristics of subjects

We analyzed 452 subjects, 46.7% women and 53.3% men with a median age of 19 (18–20) years. Based on the clinical and biochemical evaluation, SBP, DBP, serum glucose levels, total cholesterol, HDL-C, LDL-C, triglycerides, insulin, leptin and HOMA-IR were within reference limits. However, considering the 75$^{th}$ percentile, 25% of the subjects exhibited HOMA-IR levels > 3.08, 30.5% were classified as overweight, and 10.6% as obese according to their BMI (Table 1).

When classified by sex, weight, SBP, DBP, glucose and triglycerides levels were higher in men, while HDL-C, insulin and leptin levels were higher in women (Table 1). Regarding serum amino acids levels, aspartate, serine and arginine were significantly higher in women, while histidine, methionine, isoleucine, leucine, valine and the sum of BCAAs were higher in men (Table 1). When categorizing the subjects based on IR, we observed that 44.5% of the subjects presented IR. As expected, individuals with IR showed higher weight, BMI, SBP, DBP, glucose, triglycerides, insulin and leptin levels, while their HDL-C levels were lower compared to those without IR (Table 2).

Moreover, we observed that subjects with IR had higher levels of aspartate, glutamate, arginine, alanine, tyrosine, methionine, phenylalanine, lysine, isoleucine, leucine, valine and the sum of BCAAs, while glycine levels were lower than subjects without IR (Table 2).

### Genotype frequencies

We determined the genotypic frequencies of the 18 SNPs among the subjects. We found that all homozygotes with the common allele had a frequency higher than 35%, and particularly, the SNPs present in *TAT*, *OTC*, *HAL*, *BCAT2* and *BCKDH* had a frequency higher than 80%.

Regarding heterozygotes, the frequency in 15 SNPs was greater than 15%; and finally, for the homozygotes of the non-common allele (variant), 6 SNPs had a frequency greater than 10% (*GPT*, *DLD*, *SLC1A4*, *PPM1K* [rs9637599 y rs1440581], and *GCKR*). Furthermore, among the 18 evaluated SNPs, 10 SNPs were found to be in Hardy-Weinberg equilibrium (*TAT*, *HGD*, *GPT*, *HAL*, *BCAT2*, *PRODH*, *DLD*, *SLC7A9*, *PPM1K*, and *GCKR*) (Table 2 in S1 Appendix). Among these 10 SNPs, we observed that in subjects with IR, the highest frequency (85.7%) for homozygotes with the common allele was for *BCAT2*, while the highest frequency for homozygotes with the variant allele was for *GPT* (15.5%) (Table 4 in S1 Appendix).

### GRS for HOMA-IR

Among all the models analyzed in the multiple linear regression analysis (Table 5 in S1 Appendix), the model with the highest number of SNPs significantly associated with HOMA-IR included the following: rs2255543 (*HGD*), rs5747933 (*PRODH*), rs6943999 (*DLD*) and rs1007160 (*SLC7A9*) (Table 3). Subsequently, the GRS was calculated based on the standardized β coefficient and the effect size for each SNP. The GRS explained 24.6% of the HOMA-IR variability adjusted by BMI, sex and age ($R^2$ = 0.246, $p < 0.01$).

### Characteristics of subjects based on GRS

The GRS was categorized into tertiles (T1 = 149 subjects; T2 = 211 subjects; T3 = 92 subjects), revealing that 92 subjects carrying the risk alleles classified in the highest tertile (GRS-high) with a cut-off point ≥ 0.836, which had significantly higher HOMA-IR values than subjects in the first (GRS-low) and second tertiles (GRS-medium) (Fig 1). Interestingly, subjects with a high GRS showed higher levels of glucose, total cholesterol, triglycerides and insulin levels

**Table 1. Anthropometric, clinical, and biochemical characteristics of participants (n = 452).**

| Characteristic | Total sample | Men | Women | $p^1$ |
|---|---|---|---|---|
| | (n = 452) | (n = 241) | (n = 211) | |
| Age (years) | 19 (18–20) | 19 (18–20) | 19 (18–20) | - |
| Weight (kg) | 65.5 (58–75.8) | 70.5 (63–79.7) | 60 (52.5–69) | < **0.001**\* |
| BMI (kg/m$^2$) | 23.9 (21.5–26.8) | 23.8 (21.6–26.5) | 23.9 (21.3–26.9) | 0.867 |
| Systolic blood pressure (mmHg) | 110 (100–110) | 110 (100–115) | 110 (100–110) | < **0.001**\* |
| Diastolic blood pressure(mmHg) | 70 (60–80) | 70 (70–80) | 70 (60–70) | **0.001**\* |
| Glucose (mg/dL) | 79 (74–85) | 80 (76–86) | 78 (73–83) | < **0.001**\* |
| Total cholesterol (mg/dL) | 151 (132–173) | 151 (128–174) | 153 (135–173) | 0.513 |
| HDL-C (mg/dL) | 66.4 (57.6–75.3) | 64 (56.3–74.3) | 68.9 (59.1–76.9) | **0.001**\* |
| LDL-C (mg/dL) | 61.1 (44–79.7) | 61.2 (42–81) | 61 (48–79) | 0.729 |
| Triglycerides (mg/dL) | 100 (73–135) | 108 (77–147) | 92 (71–123) | **0.002**\* |
| Insulin (μU/mL) | 12 (8.94–15.7) | 11.3 (8.65–14.5) | 12.3 (9.6–16.8) | **0.007**\* |
| Leptin (ng/mL) | 11.4 (4.7–20.7) | 5.27 (3.14–9.05) | 20.4 (14.2–28.6) | < **0.001**\* |
| HOMA-IR | 2.33 (1.73–3.08) | 2.27 (1.68–2.95) | 2.34 (1.83–3.34) | 0.086 |
| Aspartate (μM) | 28.3 (22.8–34.2) | 26.5 (22–31.6) | 30.8 (24.2–37.3) | < **0.001**\* |
| Glutamate (μM) | 85.9 (71–102) | 86.4 (72–101) | 85 (69.3–103) | 0.366 |
| Serine (μM) | 127 (108–147) | 124 (108–145) | 134 (111–155) | **0.007**\* |
| Histidine (μM) | 61.8 (39.2–72.5) | 63.2 (41.5–74.7) | 60.9 (20.5–69.7) | **0.045**\* |
| Glycine (μM) | 258 (205–322) | 258 (207–322) | 256 (202–323) | 0.917 |
| Threonine (μM) | 160 (120–198) | 161 (125–197) | 159 (119–200) | 0.741 |
| Arginine (μM) | 84 (70.9–96.2) | 81.1 (69.6–94.1) | 88.7 (73.9–97.7) | **0.005**\* |
| Alanine (μM) | 557 (464–645) | 550 (463–639) | 561 (464–647) | 0.587 |
| Tyrosine (μM) | 55.1 (46.4–64.7) | 55 (46.6–65.9) | 55 (46.1–64) | 0.408 |
| Valine (μM) | 179 (136–230) | 186 (146–244) | 174 (131–220) | **0.009**\* |
| Methionine (μM) | 70.7 (43.4–99.5) | 77.2 (49–103) | 60.1 (37.5–89.9) | **0.006**\* |
| Phenylalanine (μM) | 68.4 (53.7–83.7) | 67.4 (53.3–83.2) | 69.4 (54.6–84.6) | 0.256 |
| Isoleucine (μM) | 55.2 (45.4–65.3) | 59.5 (50.1–68.3) | 52 (43.5–61) | < **0.001**\* |
| Leucine (μM) | 116 (96.5–132) | 123 (105–138) | 109 (88.6–125) | < **0.001**\* |
| Lysine (μM) | 168 (140–200) | 170 (141–203) | 167 (137–197) | 0.142 |
| Proline (μM) | 176 (123–229) | 182.5 (133–239) | 171.2 (117–220) | 0.075 |
| BCAA (μM) | 358 (290–425) | 375 (308–448) | 343 (274–391) | < **0.001**\* |

Data are shown as median (25$^{th}$ - 75$^{th}$ percentile). μM: Micromolar (μmol/L); BMI: Body Mass Index; HDL-C: High-density lipoprotein cholesterol; LDL-C: Low-density lipoprotein cholesterol; HOMA-IR: Homeostatic model assessment—insulin resistance; BCAA: Branched Chain Amino Acids.

[1]Mann-Whitney U test.

\* The difference is significant $p \leq 0.05$.

($p < 0.05$) than subjects with a low GRS (cut-off point $\leq 0.624$) without covariate adjustment. These results, except for total cholesterol, were maintained when evaluated with adjustment for age, sex and BMI (Table 4). Furthermore, subjects with a high GRS showed a positive and significant trend with higher levels in weight, BMI, glucose, total cholesterol, triglycerides, leptin, insulin and HOMA compared to subjects with medium and low GRS ($p < 0.05$) (Table 6 in S1 Appendix).

Finally, subjects with a low GRS had slightly higher arginine levels than subjects with a high GRS ($p < 0.05$) (Table 5). Some amino acids, such as proline exhibited a negative trend in their concentrations among subjects with a high GRS compared to those with a low GRS ($p < 0.05$) (Table 7 in S1 Appendix). Moreover, glycine exhibited a downward trend while

**Table 2. Anthropometric, clinical, and biochemical characteristics in subjects with or without insulin resistance (n = 452).**

| Characteristic | With IR | Without IR | $p^1$ |
|---|---|---|---|
| | (n = 201) | (n = 251) | |
| Weight (kg) | 70.5 (60.7–80) | 63 (56–71) | **< 0.001*** |
| BMI (kg/m$^2$) | 25.6 (22.4–29.2) | 22.7 (21–25) | **< 0.001*** |
| Systolic blood pressure (mmHg) | 110 (100–120) | 110 (100–110) | **< 0.001*** |
| Diastolic blood pressure (mmHg) | 70 (70–80) | 70 (60–70) | **< 0.001*** |
| Glucose (mg/dL) | 82 (77–88) | 78 (72–81) | **< 0.001*** |
| Total cholesterol (mg/dL) | 153 (135–174) | 150 (130–173) | 0.296 |
| HDL-C (mg/dL) | 62.6 (55.1–72) | 68.8 (60.3–77.2) | **< 0.001*** |
| LDL-C (mg/dL) | 64.4 (44.8–80) | 59.4 (42.8–79.5) | 0.364 |
| Triglycerides (mg/dL) | 119 (85.7–163) | 86 (67–117) | **< 0.001*** |
| Insulin (μU/mL) | 16.2 (14.1–19.9) | 9.41 (7.77–10.9) | **< 0.001*** |
| Leptin (ng/mL) | 15.8 (8.52–26.6) | 7.63 (3.66–16) | **< 0.001*** |
| Aspartate (μM) | 30.7 (24–37) | 27 (21.7–32) | **< 0.001*** |
| Glutamate (μM) | 91.4 (74.1–107) | 82.8 (69.4–94) | **< 0.001*** |
| Serine (μM) | 127 (108–146) | 128 (110–150) | 0.575 |
| Histidine (μM) | 60.4 (28.1–69.6) | 62.2 (42.4–74.3) | 0.169 |
| Glycine (μM) | 249 (196–305) | 262 (209–337) | **0.011*** |
| Threonine (μM) | 161 (120–205) | 159 (121–195) | 0.887 |
| Arginine (μM) | 87 (72–97.9) | 82 (70.6–94) | **0.045*** |
| Alanine (μM) | 586 (488–693) | 539 (446–613) | **< 0.001*** |
| Tyrosine (μM) | 59.1 (51–68.2) | 51 (44.3–61.4) | **< 0.001*** |
| Valine (μM) | 200 (150–247) | 167 (128–222) | **< 0.001*** |
| Methionine (μM) | 77.1 (46.2–103) | 61.7 (38.2–96.8) | **0.049*** |
| Phenylalanine (μM) | 74.9 (60.5–86.9) | 64.7 (51–79) | **< 0.001*** |
| Isoleucine (μM) | 56.4 (48–68.4) | 54.2 (44.8–62.8) | **0.040*** |
| Leucine (μM) | 122 (104–140) | 110 (93.7–129) | **< 0.001*** |
| Lysine (μM) | 178 (149–205) | 164 (135–191) | **0.001*** |
| Proline (μM) | 184 (124–242) | 168 (121–220) | 0.093 |
| BCAA (μM) | 381 (317–449) | 339 (278–406) | **0.008*** |

Data are shown as median (25th - 75th percentile). μM: micromolar (μmol/L); HDL-C: High-density lipoprotein cholesterol; LDL-C: Low-density lipoprotein cholesterol; BCAA: Branched Chain Amino Acids.

[1] Mann-Whitney U test.

* The difference is significant $p \leq 0.05$.

alanine and BCAA showed an upward trend, although were not statistically significant (Table 7 in S1 Appendix).

When classified by sex, we observed that woman with a low GRS had higher levels of aspartate, serine, and arginine, and lower levels of methionine, isoleucine, and leucine than men. However, women with a high GRS no longer exhibited differences in serine and methionine levels. Notably, women with a medium GRS had additionally lower levels of histidine and valine than men (Table 8 in S1 Appendix).

## Discussion

Our study shows that a GRS calculated using the number of risk alleles of the SNPs rs2255543 in *HGD*, rs5747933 in *PRODH*, rs6943999 in *DLD*, and rs1007160 in *SLC7A9* was associated

**Table 3. Genotype frequencies associated with risk of insulin resistance as assessed by HOMA-IR (n = 452).**

| Gene | Chromosome | SNP | Genotype | | | Risk allele IR[1] | β Coefficient | | $p$[2] |
|---|---|---|---|---|---|---|---|---|---|
| | | | n (%) | n (%) | n (%) | | Not Standardized ± EE | Standardized | |
| *HGD* | 3 | rs2255543 | **TT** 312 (69) | **TA** 125 (27.6) | **AA** 15 (3.31) | T | 0.25 ± 0.13 | 0.088 | 0.057[a] |
| *PRODH* | 22 | rs5747933 | **GG** 311 (68.8) | **GT** 129 (28.5) | **TT** 12 (2.65) | G | 0.34 ± 0.13 | 0.119 | **0.011*** |
| *DLD* | 7 | rs6943999 | **AA** 179 (39.6) | **AT** 217 (48) | **TT** 56 (12.4) | A | 0.21 ± 0.10 | 0.094 | **0.044*** |
| *SLC7A9* | 19 | rs1007160 | **GG** 342 (75.7) | **GT** 97 (21.5) | **TT** 13 (2.88) | G | 0.35 ± 0.14 | 0.117 | **0.012*** |

HOMA-IR: Homeostatic Model Assessment—Insulin Resistance. *HGD*: Homogentisate 1,2-Dioxygenase; *PRODH*: Proline Dehydrogenase 1; *DLD*: Dihydrolipoamide Dehydrogenase; *SLC7A9*: Solute Carrier Family 7 Member 9; SNP: Single Nucleotide Polymorphism; IR: insulin resistance; EE: Typical error.

[1]Risk allele for IR based on the HOMA-IR.

[2]The association between the SNP and HOMA-IR was obtained with a multiple linear regression model.

* The difference is significant $p \leq 0.05$.

[a]The difference is marginally significant.

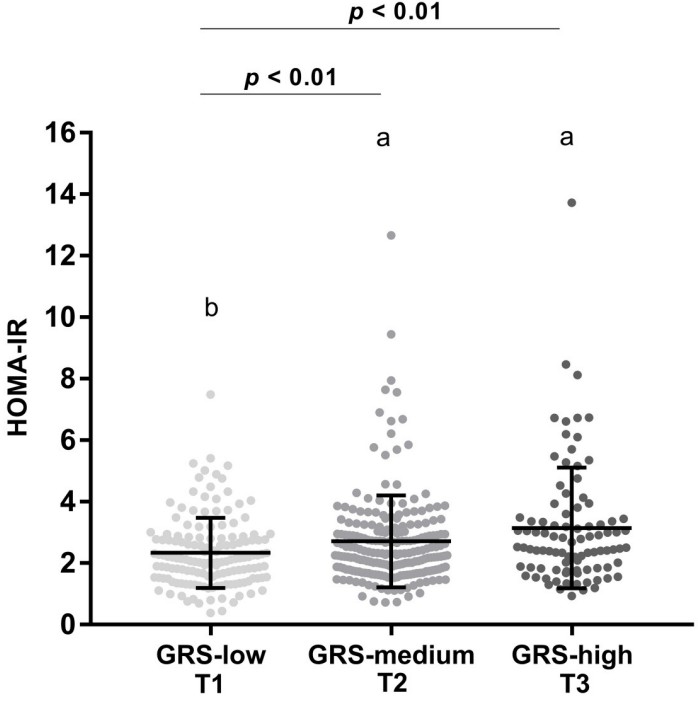

**Fig 1. Insulin resistance, quantified by the HOMA-IR (homeostatic model assessment—insulin resistance), across groups stratified into tertiles according to the genetic risk score (GRS) derived from the best model in a total of 452 subjects.** GRS-low = tertile 1 (cut-off point: 0.620); GRS-medium = tertile 2 (cut-off point: 0.742); GRS-high = tertile 3 (cut-off point: 0.836). The HOMA-IR values of subjects with a high GRS and medium GRS were significantly higher than in subjects with a low GRS. Data are shown as mean ± standard deviation. Differences are based on ANOVA adjusted for sex, age and BMI. Bonferroni´s multiple comparisons post-hoc test where groups with different letters are statistically significant, where a > b. The difference is significant $p < 0.01$.

**Table 4. Anthropometric, clinical, and biochemical parameters of subjects according to the genetic risk score for HOMA-IR (n = 452).**

| Characteristic | GRS-low | GRS-medium | GRS-high | $p^1$ | $p^2$ |
|---|---|---|---|---|---|
| | T1 (0.624) | T2 (0.742) | T3 (0.836) | | |
| | n = 149 | n = 211 | n = 92 | | |
| Weight (kg) | 66.6 ± 14 | 67.7 ± 13.9 | 69.6 ± 14.1 | 0.285 | 0.153 |
| BMI (kg/m²) | 24.1 ± 4.04 | 24.4 ± 4.39 | 25.4 ± 4.36 | 0.063 | - |
| SBP (mmHg) | 149 ± 107 | 108 ± 10.6 | 106 ± 9.95 | 0.548 | 0.269 |
| DBP (mmHg) | 69.8 ± 8.09 | 70 ± 8.30 | 70.1 ± 8.18 | 0.956 | 0.854 |
| Glucose (mg/dL) | 78.6 ± 7.56[b] | 79.6 ± 7.51[b] | 83.4 ± 19[a] | **0.004*** | **0.004*** |
| TC (mg/dL) | 150 ± 31.9[b] | 153 ± 31.2[a,b] | 161 ± 33[a] | **0.045*** | 0.095 |
| HDL-C (mg/dL) | 67.5 ± 16.9 | 67.8 ± 13.5 | 66.4 ± 15.8 | 0.594 | 0.589 |
| LDL-C (mg/dL) | 62.3 ± 28.8 | 63.2 ± 28.5 | 69.1 ± 28.3 | 0.120 | 0.188 |
| TG (mg/dL) | 109 ± 69.9[b] | 116 ± 76.8[a,b] | 133 ± 76.7[a] | **0.005*** | **0.017*** |
| Insulin (µU/mL) | 11.9 ± 5.47[b] | 13.5 ± 6.57[a] | 15.2 ± 9.50[a] | **0.002*** | **0.003*** |
| Leptin (ng/mL) | 13.3 ± 11.9 | 14.7 ± 13.1 | 17 ± 12.8 | 0.102 | 0.576 |
| HOMA-IR | 2.33 ± 1.14[b] | 2.71 ± 1.49[a] | 3.14 ± 1.96[a] | **<0.001*** | **0.001*** |

Data are shown as mean ± standard deviation. GRS: Genetic Risk Score; T1: First tertile; T2: Second tertile; T3: Third tertile; HOMA-IR: Homeostatic Model Assessment—Insulin Resistance; BMI: Body mass index; SBP: systolic blood pressure; DBP: diastolic blood pressure; TC: total cholesterol; HDL-C: High-density lipoprotein cholesterol; LDL-C: Low-density lipoprotein cholesterol; TG: Triglycerides.

[1]Differences are based on ANOVA without covariate adjustment.

[2]Differences are based on ANOVA adjusted for sex, age and BMI.

Bonferroni´s multiple comparisons post-hoc test where groups with different letters are statistically significant, where a > b.

* The difference is significant $p \leq 0.05$.

with HOMA-IR. Subjects with a high GRS had higher glucose, insulin, total cholesterol, triglycerides levels and lower arginine levels than subjects with a low GRS.

Information regarding the potential causal relationship between these SNPs and IR remains limited. At this point, we can only speculate about their implications. Homogentisate 1,2-dioxygenase (HGD) is an enzyme involved in tyrosine metabolism that converts homogentisic acid (HGA) to malate and acetoacetate [27]. Mutations in the *HGD* gene [28] lead to an autosomal recessive disorder known as alkaptonuria, which is characterized by the absence of HGD causing an accumulation of HGA [29]. Alkaptonuria is characterized by dark urine, bluish-black pigmentation in the connective tissue and arthritis [30, 31]. Moreover, a decrease in HGD activity could potentially result in decreased malate levels, thereby impacting the tricarboxylic cycle, oxidative phosphorylation, as well as amino acid and glucose metabolism, as suggested for clear cell renal carcinoma [32]. However, to our knowledge this is the first finding of a possible relationship between the rs2255543 in *HGD* and IR. Further studies are needed to understand the effect of this SNP and the activity of HGD and its consequences on IR development.

Proline dehydrogenase (PRODH) participates in the first step of proline catabolism. A previous study found that rs5747933 in *PRODH* was associated with high serum proline concentrations [33], suggesting that this SNP may decrease PRODH activity. High proline concentrations are associated with a higher incidence of T2D in the Chinese [34] and Japanese [35] adult population. Additionally, proline levels are positively correlated with IR in Mexican young adults [5]. The exact mechanism by which altered proline levels are related to T2D or IR remains unclear. However, some hypotheses could be the following: a) the increase in proline might be related to pancreatic cell dysfunction. Prolonged proline exposure increased basal insulin secretion and decreased glucose-stimulated insulin secretion in both clonal

**Table 5. Serum amino acid concentrations of subjects according to the genetic risk score for HOMA-IR (n = 452).**

| Characteristic | GRS-low | GRS-medium | GRS-high | $p^1$ | $p^2$ |
|---|---|---|---|---|---|
| | T1 (0.624) | T2 (0.742) | T3 (0.836) | | |
| | n = 149 | n = 211 | n = 92 | | |
| Aspartate (μM) | 29.6 ± 9.78 | 28.1 ± 8.63 | 30 ± 8.38 | 0.111 | 0.165 |
| Glutamate (μM) | 88.7 ± 24.6 | 85.7 ± 28.1 | 93.1 ± 30.1 | 0.056 | 0.077 |
| Serine (μM) | 132 ± 32.5 | 126 ± 30.2 | 131 ± 36.3 | 0.269 | 0.266 |
| Histidine (μM) | 56.5 ± 26.2 | 55.6 ± 27.7 | 50.9 ± 24.5 | 0.664 | 0.736 |
| Glycine (μM) | 283 ± 98.4 | 268 ± 94.5 | 258 ± 94.5 | 0.096 | 0.172 |
| Threonine (μM) | 165 ± 59.9 | 157 ± 60.5 | 167 ± 69.2 | 0.266 | 0.264 |
| Arginine (μM) | 87.3 ± 20[a] | 81.2 ± 17.1[b] | 85.1 ± 20.3[a,b] | **0.019*** | **0.017*** |
| Alanine (μM) | 543 ± 118 | 557 ± 137 | 571 ± 139 | 0.433 | 0.627 |
| Tyrosine (μM) | 55.9 ± 14.1 | 55.5 ± 15.3 | 56.6 ± 15.5 | 0.805 | 0.799 |
| Valine (μM) | 187 ± 61.5 | 187 ± 69.4 | 186 ± 60.9 | 0.927 | 0.912 |
| Methionine (μM) | 67.7 ± 34.7 | 73.7 ± 35.7 | 78.4 ± 33.1 | 0.157 | 0.093 |
| Phenylalanine (μM) | 73.4 ± 23.3 | 69.9 ± 22.5 | 73.6 ± 23.4 | 0.230 | 0.256 |
| Isoleucine (μM) | 57.6 ± 14.6 | 55.9 ± 15.8 | 57.5 ± 17 | 0.418 | 0.370 |
| Leucine (μM) | 116 ± 24.5 | 115 ± 31.4 | 117 ± 26 | 0.471 | 0.447 |
| Lysine (μM) | 173 ± 46 | 172 ± 50 | 178 ± 48.5 | 0.373 | 0.472 |
| Proline (μM) | 197 ± 88.4 | 181 ± 74 | 172 ± 70.6 | 0.134 | 0.124 |
| BCAA (μM) | 361 ± 89 | 358 ± 107 | 361 ± 90.3 | 0.705 | 0.681 |

Data are shown as mean ± standard deviation. μM: Micromolar (μmol/L); GRS: Genetic Risk Score; T1: First tertile; T2: Second tertile; T3: Third tertile; HOMA-IR: Homeostatic Model Assessment—Insulin Resistance; BCAA: Branched Chain Amino Acids.

[1] Differences are based on ANOVA without covariate adjustment.

[2] Differences are based on ANOVA adjusted for sex, age and BMI.

Bonferroni´s multiple comparisons post-hoc test where groups with different letters are statistically significant, where a > b.

* The difference is significant $p \leq 0.05$.

INS1-E insulinoma cells and isolated rat islets [36, 37]. b) Proline may function as a redox modulator. Both proline synthesis and catabolism are intricately involved in redox-active mechanisms. For instance, the catabolic activity of PRODH generates ATP and, when excessively active, leads to an elevation in reactive oxygen species (ROS) production [37]. Several studies have linked the production of ROS to IR [38–40]. c) The modulation of glutamate production can impact glucagon secretion. Proline oxidation results in glutamate production, which in turn induces glucagon release in pancreatic alpha cells [41, 42]. Additionally, glutamate facilitates the conversion of pyruvate to alanine. Glucagon secretion stimulates hepatic gluconeogenesis, while the high availability of alanine serves as a gluconeogenic substrate, potentially amplifying this metabolic pathway [34].

Regarding the SNP rs6943999, it is located in the promoter region of the *DLD* gene. Dihydrolipoamide dehydrogenase (DLD) is an enzyme that catalyzes the oxidation of NADH to $NAD^+$ in the glycine cleavage system. Moreover, DLD is the E3 component of three multienzyme dehydrogenase complexes (pyruvate, alpha-ketoglutaramate, and BCKDH complex). The BCKDH complex modulates BCAA catabolism. Subjects with IR and obesity have increased serum BCAAs levels [43], which could be due to both a decrease in the expression of BCAA catabolic enzymes or a decrease in its activity [44]. The increase in BCAAs may cause the activation of the mammalian target of rapamycin (mTOR) pathway, subsequently activating downstream kinases such as p70S6 ribosomal kinase (p70S6K or S6K). This kinase can

phosphorylate insulin receptor substrate (IRS-1), potentially leading to the suppression of insulin signaling on serine/threonine residues [45, 46].

Furthermore, DLD also conforms the pyruvate dehydrogenase complex, implying that a decrease in DLD expression might be associated to an accumulation of pyruvate, a gluconeogenic substrate, particularly during prolonged fasting. An excess of pyruvate would increase gluconeogenesis [47]. That said, further research is needed to determine whether the presence of rs6943999 affects *DLD* expression altering BCAA and pyruvate homeostasis, and thus, IR.

*SLC7A9* encodes for a sodium-independent cationic amino acids transporter, which is primarily responsible for the uptake of certain amino acids, such as cystine, lysine, arginine and neutral amino acids [48]. However, to our knowledge, there are no studies reporting an association between rs1007160 and IR. Lower expression of amino acid transporters, including SLC7A9, has been observed in hepatocytes from mice with diet-induced obesity, and this decrease was associated with hepatic steatosis, hyperlipidemia, obesity, and IR [49, 50]. As a non-synonymous SNP, rs1007160 could potentially impact the structure or function of the transporter [51], resulting in a lower uptake of amino acids such as arginine and potentially affecting regulatory mechanisms. For example, arginine is the main substrate for nitric oxide synthesis. Through this pathway, arginine modulates glucose and lipid oxidation, and insulin sensitivity [52]. In addition, arginine can also activate the mTOR signaling mechanism, promoting protein synthesis and cell growth [53].

Concerning the differences on amino acids levels observed between men and women, our results are consistent with previous reports demonstrating that men have higher histidine, methionine, tyrosine and BCAA concentrations than women [54–56]. A possible explanation to the lower concentration of BCAAs in women could be related to the high catabolism of BCAAs in adipocytes [57], and the higher amount of body fat present in women [58]. Moreover, methionine has been positively associated with adiposity; in fact, methionine restriction is thought to improve insulin sensitivity and increase weight loss in humans and mice [59, 60]. Interestingly, BCAA levels were lower in women, regardless of the GRS tertile. However, the differences observed in serine and methionine, when classified by sex in subjects with low GRS, were no longer present in subjects with high GRS. This suggests that the presence of SNPs may have a sex-dependent effect on certain amino acid catabolism, influencing their plasma concentrations, which requires further research for elucidation.

Our study has several strengths. Firstly, it stands as one of the initial studies to evaluate various SNPs in genes associated with amino acid metabolism, and their relationship with IR risk in a population of young Mexican adults. Secondly, these results provide evidence of novel SNPs linked to IR, along with the identification of amino acids as potential biomarkers for cardiometabolic risk. Thirdly, the implementation of the GRS might facilitate the early identification of young subjects at increased risk of IR. This approach, involving the evaluation of diverse SNPs, could have a greater clinical impact than the assessment of a single SNP alone.

While our findings must be validated in an independent population and should include an evaluation of the effect of the nutritional conditions of the subjects on their plasma amino acid levels, subjects with higher GRS may benefit from preventive lifestyle interventions and/or pharmacological treatment to reduce obesity to prevent the development of IR. Moreover, another limitation of our study lies in its cross-sectional design, which precludes to determine the causality of the results. Further research is required to evaluate whether these SNPs indeed harbor a causal relationship with the development of IR over a time interval, and to determinate how they impact amino acid concentration. In addition, the study focused on a specific population of young adults, which limits the generalizability of the findings to other age groups or populations. While the power analysis of the utilized sample size exceeded 80%, which is

considered acceptable, further research with diverse cohorts would be valuable to validate the observed associations.

In conclusion, we calculated a GRS using the number of risk alleles of the SNPs in *HGD*, *PRODH*, *DLD* and *SLC7A9* genes. Subjects with high GRS had higher levels of HOMA-IR, glucose, insulin, total cholesterol and triglycerides, and lower levels of arginine than subjects with low GRS. The application of a GRS based on variant of genes associated with amino acid metabolism may be useful for the early identification of subjects at increased risk of IR.

## Supporting information

**S1 Appendix. Additional tables.**
(DOCX)

## Acknowledgments

The authors would like to thank all the members of the Departamento Fisiología de la Nutrición, as well as the staff of the Laboratorio Clínico de la Facultad de Ciencias Quimicas and all the participants that supported this effort.

## Author Contributions

**Conceptualization:** Eunice Lares-Villaseñor, Martha Guevara-Cruz, Lilia G. Noriega, Juan M. Vargas-Morales.

**Formal analysis:** Eunice Lares-Villaseñor, Martha Guevara-Cruz, Omar Granados-Portillo, Isabel Medina-Vera, Luis E. González-Salazar, Adriana M. López-Barradas, Azalia Avila-Nava, Patricia E. Cossío-Torres, Lilia G. Noriega, Juan M. Vargas-Morales.

**Funding acquisition:** Martha Guevara-Cruz, Lilia G. Noriega.

**Investigation:** Eunice Lares-Villaseñor, Martha Guevara-Cruz, Samuel Salazar-García, Omar Granados-Portillo, Mariela Vega-Cárdenas, Miguel Ernesto Martinez-Leija, Isabel Medina-Vera, Luis E. González-Salazar, Liliana Arteaga-Sanchez, Rocío Guízar-Heredia, Karla G. Hernández-Gómez, Aurora E. Serralde-Zúñiga, Edgar Pichardo-Ontiveros, Adriana M. López-Barradas, Laura Guevara-Pedraza, Guillermo Ordaz-Nava, Azalia Avila-Nava, Ulises de la Cruz-Mosso.

**Methodology:** Miguel Ernesto Martinez-Leija, Juan M. Vargas-Morales.

**Project administration:** Martha Guevara-Cruz, Lilia G. Noriega, Juan M. Vargas-Morales.

**Resources:** Armando R. Tovar, Ulises de la Cruz-Mosso, Celia Aradillas-García, Diana P. Portales-Pérez, Juan M. Vargas-Morales.

**Supervision:** Martha Guevara-Cruz, Samuel Salazar-García, Aurora E. Serralde-Zúñiga, Armando R. Tovar, Patricia E. Cossío-Torres, Celia Aradillas-García, Diana P. Portales-Pérez, Lilia G. Noriega, Juan M. Vargas-Morales.

**Writing – original draft:** Eunice Lares-Villaseñor.

**Writing – review & editing:** Martha Guevara-Cruz, Samuel Salazar-García, Omar Granados-Portillo, Miguel Ernesto Martinez-Leija, Isabel Medina-Vera, Luis E. González-Salazar, Rocío Guízar-Heredia, Azalia Avila-Nava, Patricia E. Cossío-Torres, Celia Aradillas-García, Diana P. Portales-Pérez, Lilia G. Noriega, Juan M. Vargas-Morales.

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
