## [Decision Letter · Decision Letter 0]

8 Nov 2023

PONE-D-23-27177Genetic risk score for insulin resistance based on gene variants associated to amino acid metabolism in young adultsPLOS ONE

Dear Dr. Noriega,

Thank you for submitting your manuscript to PLOS ONE. After careful consideration, we feel that it has merit but does not fully meet PLOS ONE’s publication criteria as it currently stands. Therefore, we invite you to submit a revised version of the manuscript that addresses the points raised during the review process.

We look forward to receiving your revised manuscript.

Kind regards,

Hongsong Zhang

Academic Editor

PLOS ONE

Journal Requirements:

This work was supported by CONACYT-202721 and CONACYT-CF2019-2096049 to LGN, by CONACYT-PN-2016-01-3324 to MGC, and by consultancy and industrial services 17 and 34 to JMVM from Facultad de Ciencias Químicas from the Universidad Autónoma de San Luis Potosí.

Reviewers' comments:

Reviewer's Responses to Questions

**Comments to the Author**

1. Is the manuscript technically sound, and do the data support the conclusions?

Reviewer #1: Yes

Reviewer #2: Partly

Reviewer #3: No

2. Has the statistical analysis been performed appropriately and rigorously? 

Reviewer #1: I Don't Know

Reviewer #2: Yes

Reviewer #3: No

3. Have the authors made all data underlying the findings in their manuscript fully available?

Reviewer #1: Yes

Reviewer #2: Yes

Reviewer #3: No

4. Is the manuscript presented in an intelligible fashion and written in standard English?

Reviewer #1: Yes

Reviewer #2: Yes

Reviewer #3: Yes

5. Review Comments to the Author

Reviewer #1: The paper focused on the Genetic risk score for insulin resistance based on gene variants associated with amino acid metabolism in young adults and the results were impressive. The authors have provided detailed information and presented their research clearly and concisely, making it easy to follow along with their findings. There are Minor concerns that need to be answered or modified:

1- Are diabetic patients excluded? The paper only mentions that participants using hypoglycemic agents were not included. However, it is not clear whether any newly diagnosed diabetic patients during the study were included in the study.

2- For GRS calculation, the cut used for HOMA-IR is >2.5 ?

3- The unit of amino acids in Table 2, is Mmol. Does it mean Mmol/L ?

4- Given the difference in amino acid levels between genders, it is recommended to present the results of Table 5, for each gender separately.

5- Did the study take into account the effect of nutritional conditions on amino acid levels? If not, it may be worth noting this as a limitation in the study's limitations section.

Reviewer #2: Eunice Lares-Villaseñor and his/her colleagues developed and evaluated Genetic Risk Score (GRS) for insulin resistance in young adults based on SNPs associated to amino acid metabolism in young Mexican adults. The results of this study, with its own limitations, could help for early identification of subjects at increased risk of insulin resistance for young Mexican adults.

There are some concerns:

1.In the last paragraph of the introduction, ............to develop a GRS to predict the risk of IR in young Mexican adults based on the determination of SNPs of ......... . Since this study didn’t incorporate all SNPs that are related to amino acid metabolism in the construction of GRS, it would be nice to say ‘’some selected SNPs’’ or otherwise better to mention the exact number of SNPs used in the construction of GRS.

2.It is not clear how the study subjects were recruited to be included in this study. Are they selected randomly? Or what? Were subjects recruited from general population? T2D population? Or normal population? Please briefly describe about recruitment of study populations.

3. What was the reason behind selecting body mass index (BMI) >20 and <40kg/m2 ? please justify why you choose it.

4.This study used previous study conducted in Turkey and assessed IR in women with polycystic ovary syndrome to establish HOMA-IR. I have few questions regarding this. Why you choose this study? Wasn’t there previous study in Mexico or any other study in the Latino populations? Why you choose IR cutoff point in patient with polycystic ovary syndrome? by using this cutoff point, the result of this study findings could be misleading. Please reconsider it.

5.In the SNP selection criteria, you used only one criterion which is frequency > 10%? Didn’t employed any other criteria?

6.The paragraph which described about genotyping is not clear. Could you please refine it?

7.Kolmogorov-Smirnov Z Test was employed to analyze the continuous variable distribution and Mann-Whitney U test was used to compare the difference between in anthropometric, clinical, and biochemical variables based on sex only if the continuous variable is found not to be normally distributed. What if it were normally distributed?

8.In the statistical analysis, backward-stepwise method was used to determine the final model. Why backward-stepwise was selected over other methods? Please justify it.

9.This study categorized the GRS into tertiles. It would be nice if you describe how many subjects were included in each tertiles. Readers may be interested to know how many subjects there in each tertiles.

10.Did this study check for multiple testing? Because many SNPs were used to construct GRS.

11.In table 1, there is a huge difference in Leptin concentration between male and female. Please check it. In the same table, the weight of total subjects 70.5 (63-79.7) and male subjects (70.5 (63-79.7) are exactly similar, however the weight of female subjects 60 (52.5-69) is far less. How could this happen? Could you please check it?

12.Did this study evaluated the effect allele/genotype frequency between subjects with and without insulin resistant? Readers might be interested seeing the results.

13.In table 4, the figures are ordered numbers from first tertile through second tertile and third tertile. So that what if you test the trend between first, second and third tertiles using Jonckheere-Terpstra trend test (Distribution-Free k-Sample Test Against Ordered Alternatives) (Jonckheere, A. R. “A Distribution-Free k-Sample Test Against Ordered Alternatives.” Biometrika 41, no. 1/2 (1954): 133–45. https://doi.org/10.2307/2333011.) than Kruskal-Wallis? The same also goes to table 5.

14.The last figure is not clear. Even it didn’t have a description about the figure. Please check it.

Reviewer #3: The manuscript titled "Genetic Risk Score for Insulin Resistance Based on Gene Variants Associated with Amino Acid Metabolism in Young Adults" presents an investigation into the development of a Genetic Risk Score (GRS) for insulin resistance in young adults, utilizing single nucleotide polymorphisms (SNPs) associated with amino acid metabolism. The study aims to identify a GRS that could aid in the early identification of individuals at risk of insulin resistance. The authors conclude that utilizing a GRS based on variants within genes associated with amino acid metabolism may be beneficial for the early identification of individuals at an increased risk of insulin resistance. While the study provides interesting insights into the potential role of genetic variants in amino acid metabolism and their association with insulin resistance, there are some limitations that impede the acceptance of the manuscript for publication.

Observations:

To the best of my knowledge, there is no consensus in the literature regarding the association between circulating amino acids and insulin resistance.

The manuscript suggests that SNPs within genes related to amino acid metabolism may contribute to increased levels of specific amino acids, potentially leading to insulin resistance. However, the authors fail to demonstrate a clear relationship between amino acid concentration and insulin resistance.

It is important for the authors to clarify why they believe this particular population is the best choice for calculating insulin resistance. Are there any specific risk factors that justify this choice? The choice of age range (18-25 years) should be better justified, considering the higher prevalence of insulin resistance in older individuals.

Methods:

The study would benefit from providing more detailed data that would allow other researchers to replicate the study effectively. This would enhance the credibility and scientific value of the findings.

The study employed a cross-sectional design with a sample size of 452 subjects aged over 18. Have the authors performed any sample size power calculation?

The analysis should be adjusted according to sex and BMI, as obesity is related to insulin resistance.

The abstract mentions the use of 18 SNPs for GRS construction, but only 10 were in Hardy-Weinberg equilibrium and included in the models for GRS construction. Finally, only four were used in the association analysis. This should be clearly stated in the abstract to avoid confusion.

Risk allele information:

In the supplementary material, it is unclear which allele was referred to as the risk allele in the GRS calculation in the missing information lines.

In Supplementary Table 1, there is an ambiguous reference for the risk allele. The authors should clarify the intended meaning of the column headings: "PRESUMED REFERENCE RISK ALLELE."

How many risk alleles do individuals with higher GRS have? This information is necessary to be useful in clinical practice and replication studies.

Discussion:

There is a lack of discussion about the association between GRS and amino acid concentration, which is a primary aspect of the manuscript's hypothesis. In my opinion, this manuscript lacks sufficient scientific support to affirm that circulating amino acids are a risk factor for insulin resistance.

Additional considerations:

Provide reference values for blood pressure.

Use correct abbreviations for cholesterol in lipoprotein particles (e.g., HDL-C, LDL-C).

Include the number of risk alleles individuals with higher GRS have.

As a limitation, it is important to highlight that the study focused on a specific population of young adults, which may limit the generalizability of the findings to other age groups or populations. Further investigations involving diverse cohorts would be valuable in confirming the observed associations.

6. PLOS authors have the option to publish the peer review history of their article (what does this mean?). If published, this will include your full peer review and any attached files.

Reviewer #1: No

Reviewer #2: **Yes: **Dr. Nardos Abebe, PhD

Reviewer #3: No

---

## [Author Response · Author response to Decision Letter 0]

11 Jan 2024

Answers to Reviewers’ comments:

We thank the reviewers for their time revising our manuscript and for their valuable suggestions that have enabled to improve the manuscript. We sincerely appreciate their positive comments. A detailed point-by-point answer to their comments is provided below.

Reviewer #1: 

The paper focused on the Genetic risk score for insulin resistance based on gene variants associated with amino acid metabolism in young adults and the results were impressive. The authors have provided detailed information and presented their research clearly and concisely, making it easy to follow along with their findings. There are Minor concerns that need to be answered or modified:

1. Are diabetic patients excluded? The paper only mentions that participants using hypoglycemic agents were not included. However, it is not clear whether any newly diagnosed diabetic patients during the study were included in the study.

A. Thank you for your observation. Diabetic patients were excluded, and we have now specified this in the exclusion criteria in the Methods section.

“The exclusion criteria included pregnancy, substance abuse, history of cardiovascular events, chronic diseases (including individuals previously and newly diagnosed with T2D), and treatment with hypoglycemic, antihypertensive agents, agents used to treat dyslipidemias, steroids, and immunosuppressors”.

2. For GRS calculation, the cut used for HOMA-IR is >2.5?

A. Thank you for your observation. We have now specified the cut-off point used for HOMA-IR in the GRS calculation section within the methods.

“The GRS was calculated for each individual as the sum of the number of IR risk alleles based on the highest HOMA-IR value (preference cut-off ≥ 2.5)”.

3. The unit of amino acids in Table 2, is Mmol. Does it mean Mmol/L?

A. Thank you for your observation. The units are expressed in micromolar, which are micromol/liter (μmol/L). We have now specified this information in the table footer. 

4. Given the difference in amino acid levels between genders, it is recommended to present the results of Table 5, for each gender separately.

A. Thank you for your observation. We have now included the suggested table in the supplementary material (Supplementary Table 8) and described its content in the Results and Discussion sections. 

Results section:

“When classified by sex, we observed that woman with a low GRS had higher levels of aspartate, serine, and arginine, and lower levels of methionine, isoleucine, and leucine than men. However, women with a high GRS no longer exhibited differences in serine and methionine levels. Notably, women with a medium GRS had additionally lower levels of histidine and valine than men (Table 8 in S1 Appendix).” 

Discussion section:

“Interestingly, BCAA levels were lower in women, regardless of the GRS tertile. However, the differences observed in serine and methionine, when classified by sex in subjects with low GRS, were no longer present in subjects with high GRS. This suggests that the presence of SNPs may have a sex-dependent effect on certain amino acid catabolism, influencing their plasma concentrations, which requires further research for elucidation.”

5. Did the study take into account the effect of nutritional conditions on amino acid levels? If not, it may be worth noting this as a limitation in the study's limitations section.

A. Thank you for your observation. Unfortunately, we didn´t evaluate the effect of the nutritional conditions of the individuals on the amino acid levels. We have now address this issue as a limitation of our study.

…“While our findings must be validated in an independent population and should include an evaluation of the effect of the nutritional conditions of the subjects on their plasma amino acid levels…”

Reviewer #2

Eunice Lares-Villaseñor and his/her colleagues developed and evaluated Genetic Risk Score (GRS) for insulin resistance in young adults based on SNPs associated to amino acid metabolism in young Mexican adults. The results of this study, with its own limitations, could help for early identification of subjects at increased risk of insulin resistance for young Mexican adults.

There are some concerns:

1. In the last paragraph of the introduction, ............to develop a GRS to predict the risk of IR in young Mexican adults based on the determination of SNPs of ......... . Since this study didn’t incorporate all SNPs that are related to amino acid metabolism in the construction of GRS, it would be nice to say ‘’some selected SNPs’’ or otherwise better to mention the exact number of SNPs used in the construction of GRS.

A. Thank you for your observation. We have now modified the last paragraph as suggested.

…“Therefore, the aim of this study was to develop a GRS to predict the risk of IR in young Mexican adults based on the determination of some selected SNPs of genes related to the metabolism of amino acids”.

2. It is not clear how the study subjects were recruited to be included in this study. Are they selected randomly? Or what? Were subjects recruited from general population? T2D population? Or normal population? Please briefly describe about recruitment of study populations.

A. Thank you for your observation. Subjects were invited to participate in the study during university admission procedures. We have now specified this in the study population section in methods. Subjects previously and newly diagnosed with T2D were excluded from the study.

“We carried out a cross-sectional study. Subjects from the general population who were carrying out university admission procedures were invited to participate in the study.” 

3. What was the reason behind selecting body mass index (BMI) >20 and <40kg/m2? please justify why you choose it.

A. Thank you for your observation. In fact, we included subjects with a BMI between 18.5 kg/m2 and 40 kg/m2. This has been corrected in the manuscript, and we apologize for our mistake. We selected subject with up to 40 kg/m2 to include patients with normal weight, as well as those with grade 1 and 2 obesity. Grade 3 obesity was excluded due to the reported exacerbated degree of inflammation reported in those subjects [1-3]. 

“The participants were Mexicans from 18 to 25 years old with body mass index (BMI) ≥ 18.5 and < 40 kg/m2.”

References:

 El-Mikkawy, D.M.E., EL-Sadek, M.A., EL-Badawy, M.A. et al. Circulating level of interleukin-6 in relation to body mass indices and lipid profile in Egyptian adults with overweight and obesity. Egypt Rheumatol Rehabil 47, 7 (2020). https://doi.org/10.1186/s43166-020-00003-8

 Khaodhiar L, Ling PR, Blackburn GL, Bistrian BR. Serum levels of interleukin-6 and C-reactive protein correlate with body mass index across the broad range of obesity. JPEN J Parenter Enteral Nutr. 28, 6 (2004). https://doi.org/10.1177/0148607104028006410. 

 Cohen E, Margalit I, Shochat T, Goldberg E, Krause I. Markers of Chronic Inflammation in Overweight and Obese Individuals and the Role of Gender: A Cross-Sectional Study of a Large Cohort. J Inflamm Res. 25, 14 (2021). https://doi.org/10.2147/JIR.S294368. 

4. This study used previous study conducted in Turkey and assessed IR in women with polycystic ovary syndrome to establish HOMA-IR. I have few questions regarding this. Why you choose this study? Wasn’t there previous study in Mexico or any other study in the Latino populations? Why you choose IR cutoff point in patient with polycystic ovary syndrome? by using this cutoff point, the result of this study findings could be misleading. Please reconsider it.

A. Thank you for your observation. Previous studies performed in our country with the Mexican population have reported a HOMA-IR cutoff point of 2.5. We apologize for the omission of these references that lead to this confusion. We have now updated the references in the manuscript [1-3].

References:

 Arellano-Campos, O., Gómez-Velasco, D.V., Bello-Chavolla, O.Y. et al. Development and validation of a predictive model for incident type 2 diabetes in middle-aged Mexican adults: the metabolic syndrome cohort. BMC Endocr Disord 19, 41 (2019). https://doi.org/10.1186/s12902-019-0361-8

 Gómez-García A, Nieto-Alcantar E, Gómez-Alonso C, Figueroa-Nuñez B, Alvarez-Aguilar C. Anthropometric parameters as predictors of insulin resistance in overweight and obese adults. Aten Primaria. 42,7 (2010). https://doi.org/10.1016/j.aprim.2009.10.015.

 Aguilar-Salinas CA, Olaiz G, Valles V, Torres JM, Gómez Pérez FJ, Rull JA, Rojas R, Franco A, Sepulveda J. High prevalence of low HDL cholesterol concentrations and mixed hyperlipidemia in a Mexican nationwide survey. J Lipid Res. 42, 8 (2001). 

5. In the SNP selection criteria, you used only one criterion which is frequency > 10%? Didn’t employed any other criteria?

A. Thank you for your observation. Another criterion considered in the selection of SNPs was the previous association of SNPs present in genes coding for enzymes related to amino acid metabolism with alterations in plasma amino acid concentration and/or with cardiometabolic risk factors. Additionally, we preferentially selected non-synonymous SNPs. We have no added these additional criteria to the methods section.

“We performed a bibliographic search to identify SNPs present related to amino acid metabolism (such as catabolic enzymes or amino acid transporters), which were previously associated with alterations in plasma amino acids concentration and/or with cardiometabolic risk factors. The SNPs were selected when a frequency > 10% was reported for the Mexican or Latino population and using data managers such as… Additionally, non-synonymous SNPs were preferentially selected.”

6. The paragraph which described about genotyping is not clear. Could you please refine it?

A. Thank you for your observation. We have modified the genotyping section in methods. 

“These 18 SNPs were analyzed using allelic discrimination assays using TaqMan probes (AppliedBiosystems®) by the real time polymerase chain reaction (RT-PCR) on a LightCycler® 480 instrument (Roche®). Briefly, a master mixture was prepared considering for each sample 0.75 µL of TaqMan probe, 0.25 µL of molecular grade nuclease-free ultrapure water (USB®, USA), and 5 µL of Probes Master (LightCycler® 480), following the manufacturer's instructions. Then, to perform PCR, 4 µL of previously adjusted DNA and 6 µL of the master mixture were added to each well of the 96-well plates (Roche®). Negative controls were also included, which only carried the master mixture and nuclease-free water. The reactions were performed in duplicate. The cycling conditions consisted of an initial pre-incubation cycle at 95 °C for 10 min, followed by 45 cycles of denaturation at 95 °C for 12 s, annealing at 60 °C for 50 s and extension at 72 ° C for 2 s and a cooling cycle at 40 °C for 30 s. For allelic discrimination results, the context sequence for each Taqman probe and the fluorophores targeting each allele were previously verified based on information reported by the manufacturer.” 

7. Kolmogorov-Smirnov Z Test was employed to analyze the continuous variable distribution and Mann-Whitney U test was used to compare the difference between in anthropometric, clinical, and biochemical variables based on sex only if the continuous variable is found not to be normally distributed. What if it were normally distributed?

A. Thank you for your observation. We have now specified in statistical analyses of the methods section the tests used for normally distributed variables.

“The Student T test was used for variables with a parametric distribution, while the Mann-Whitney U test for non-parametric variables to analyze differences in anthropometric, clinical, and biochemical data.”

8. In the statistical analysis, backward-stepwise method was used to determine the final model. Why backward-stepwise was selected over other methods? Please justify it. 

A. Thank you for your question. We chose the backward-stepwise method because it aligns with the nature of our hypothesis and research objectives. This method is effective in preventig model overfitting by progressively eliminating less relevant SNPs that did not contribute significantly to the model. Also, the progressive elimination of less relevant SNPs helps to avoid collinearity problems and improves the stability of the model to generate the GRS. In summary, we opted for the backward-stepwise method because of its ability to improve model efficiency, avoid overfitting, simplify interpretation, and fit the particular characteristics of our data set and research objectives.

9. This study categorized the GRS into tertiles. It would be nice if you describe how many subjects were included in each tertiles. Readers may be interested to know how many subjects there in each tertiles.

A. Thank you for your observation. We have now specified how many subjects were in each tertile in the results section.

“The GRS was categorized into tertiles (T1 = 149 subjects; T2= 211 subjects; T3= 92 subjects),”

10. Did this study check for multiple testing? Because many SNPs were used to construct GRS. 

A. Thank you for your observation. We first assessed the effect of each SNP on the HOMA-IR variable using a general linear model adjusted for age, sex and BMI. Subsequently, we analyzed the 10 SNPs collectively in multiple linear regression models, considering the statistical method´s assumptions and employing both backward and forward methods. We selected the model with the highest number of significant SNPs to construct the GRS (only 4 SNPs), using the adjusted β of the model. We further examined the association between the obtained GRS scores and the HOMA-IR variable adjusting for sex, age, and BMI. Finally, we conducted a GRS stratification by tertiles, analyzing each biochemical, clinical and anthropometric variable through ANOVA and Bonferroni post-hoc tests. We have now completed this information in statistical analyses of the methods section. 

“For GRS, the effect of each SNP on the HOMA-IR variable was first asssessed using a general linear model adjusted for age, sex and BMI. Then, multiple linear regression analysis was used to assess the association between HOMA-IR (dependent variable) and the 10 SNPs (independent variables). Non-collinearity was previously evaluated between the independent variables. The backward-stepwise method was used to select the final model. Significant SNPs were used for the GRS. Moreover, we evaluated the association between the obtained GRS and the HOMA-IR variable adjusting for age, sex, and BMI using a generalized linear model. Subsequently, the GRS was categorized into tertiles. This categorization was used to assess the trends in each anthropometric, clinical, and biochemical variable among the subjects using the Jonckheere-Terpstra test. Lastly, ANOVA and Bonferroni post-hoc test with and without adjustment for covariates (age, BMI and sex) were used to assess differences in the variables of interest and the GRS. Previously, the nonparametric data were logarithmically transformed.”

11. In table 1, there is a huge difference in Leptin concentration between male and female. Please check it. In the same table, the weight of total subjects 70.5 (63-79.7) and male subjects (70.5 (63-79.7) are exactly similar, however the weight of female subjects 60 (52.5-69) is far less. How could this happen? Could you please check it?

A. Thank you for your observation. We have carefully reviewed the data from table 1 and identifyed duplicated weight data. This has been now corrected in the table, and we apologize for any confusion caused by our mistake. Additionally, concerning the leptin data, we confirmed that women have higher levels of leptin than men. Our findings align with studies previously conducted in a population similar to ours [1,2]. Probably, this could be explained by the higher fat mass in women.

References

 García-Jiménez S, Bernal Fernández G, Martínez Salazar MF, Monroy Noyola A, Toledano Jaimes C, Meneses Acosta A, Gonzalez Maya L, Aveleyra Ojeda E, Terrazas Meraz MA, Boll MC, Sánchez-Alemán MA. Serum leptin is associated with metabolic syndrome in obese Mexican subjects. J Clin Lab Anal. 29, 2 (2015). https://doi.org/10.1002/jcla.21718. 

 López-Quintero A, García-Zapién AG, Flores-Martínez SE, Díaz-Burke Y, González-Sandoval CE, Lopez-Roa RI, Medina-Díaz E, Muñoz-Almaguer ML, Sánchez-Corona J. Contribution of polymorphisms in the LEP, LEPR and RETN genes on serum leptin and resistin levels in young adults from Mexico. Cell Mol Biol (Noisy-le-grand). 30, 63 (2017). https://doi.org/10.14715/cmb/2017.63.8.3. PMID: 28886308.

12. Did this study evaluated the effect allele/genotype frequency between subjects with and without insulin resistant? Readers might be interested seeing the results.

A. Thank you for your suggestion. We have added a new table containing this information in the supplementary material (Supplementary Table 4).

“Among these 10 SNPs, we observed that in subjects with IR, the highest frequency (85.7%) for homozygotes with the common allele was for BCAT2, while the highest frequency for homozygotes with the variant allele was for GPT (15.5%) (Table 4 in S1 Appendix).”

13. In table 4, the figures are ordered numbers from first tertile through second tertile and third tertile. So that what if you test the trend between first, second and third tertiles using Jonckheere-Terpstra trend test (Distribution-Free k-Sample Test Against Ordered Alternatives) (Jonckheere, A. R. “A Distribution-Free k-Sample Test Against Ordered Alternatives.” Biometrika 41, no. 1/2 (1954): 133–45. https://doi.org/10.2307/2333011.) than Kruskal-Wallis? The same also goes to table 5. 

A. Thank you for your suggestion. We have incorporated the recommended analysis using the Jonckheere-Terpstra trend test. The results of this analysis have been included in the supplementary material (Supplementary Table 6 and 7) and are now described in the Results section.

“Furthermore, subjects with a high GRS showed a positive and significant trend with higher levels in weight, BMI, glucose, total cholesterol, triglycerides, leptin, insulin and HOMA compared to subjects with medium and low GRS (p < 0.05) (Table 6 in S1 Appendix). 

Finally, subjects with a low GRS had slightly higher arginine levels than subjects with a high GRS (p < 0.05) (Table 5). Some amino acids, such as proline exhibited a negative trend in their concentrations among subjects with a high GRS compared to those with a low GRS (p<0.05) (Table 7 in S1 Appendix). Moreover, glycine exhibited a downward trend, while alanine and BCAA showed an upward trend, although they were not statistically significant (Table 7 in S1 Appendix).”

14. The last figure is not clear. Even it didn’t have a description about the figure. Please check it.

A. Thank you for your observation. We apologize for the lack of clarity in the last figure. We have now improved the description of figure 1. 

Figure legend

Figure 1. Insulin resistance, quantified by the HOMA-IR (homeostatic model assessment - insulin resistance), across groups stratified into tertiles according to the genetic risk score (GRS) derived from the best model in a total of 452 subjects. GRS-low = tertile 1 (cut-off point: 0.620); GRS-medium = tertile 2 (cut-off point: 0.742); GRS-high = tertile 3 (cut-off point: 0.836). The HOMA-IR values of subjects with a high GRS and GRS-medium were significantly higher than in subjects with a low GRS. 

Data are shown as mean ± standard deviation. 

Differences are based on ANOVA adjusted for sex, age and BMI.

Bonferroni´s multiple comparisons post-hoc test where groups with different letters are statistically significant, where a > b.

The difference is significant p < 0.01.

Reviewer #3:

The manuscript titled "Genetic Risk Score for Insulin Resistance Based on Gene Variants Associated with Amino Acid Metabolism in Young Adults" presents an investigation into the development of a Genetic Risk Score (GRS) for insulin resistance in young adults, utilizing single nucleotide polymorphisms (SNPs) associated with amino acid metabolism. The study aims to identify a GRS that could aid in the early identification of individuals at risk of insulin resistance. The authors conclude that utilizing a GRS based on variants within genes associated with amino acid metabolism may be beneficial for the early identification of individuals at an increased risk of insulin resistance. While the study provides interesting insights into the potential role of genetic variants in amino acid metabolism and their association with insulin resistance, there are some limitations that impede the acceptance of the manuscript for publication.

Observations:

1. To the best of my knowledge, there is no consensus in the literature regarding the association between circulating amino acids and insulin resistance. 

A. Thank you for your observation. There is indeed a controversy in the literature regarding whether circulating amino acid levels are altered due to insulin resistance or whether insulin resistance induces changes in the circulating amino acid profile. To better understand the direction of causality and the intricate relationship between amino acids and insulin resistance requires further research, which falls beyond the scope of this manuscript.

2. The manuscript suggests that SNPs within genes related to amino acid metabolism may contribute to increased levels of specific amino acids, potentially leading to insulin resistance. However, the authors fail to demonstrate a clear relationship between amino acid concentration and insulin resistance. 

A. Thank you for your observation. In Table 2, we have presented how amino acids are modified in subjects with and without insulin resistance. It is observed that the majority of amino acids increase in subjects with insulin resistance. The association between amino acid profiles and insulin resistance has been observed in previous studies [1-2], and we have also identified both positive and negative correlations between specific amino acids and insulin resistance in our previous work [3].

References:

 Chen S, Miki T, Fukunaga A, Eguchi M, Kochi T, Nanri A, et al. Associations of serum amino acids with insulin resistance among people with and without overweight or obesity: A prospective study in Japan. Clin Nutr 2022; 41:1827–33. https://doi.org/10.1016/J.CLNU.2022.06.039.

 Palmer ND, Stevens RD, Antinozzi PA, Anderson A, Bergman RN, Wagenknecht LE, et al. Metabolomic profile associated with insulin resistance and conversion to diabetes in the Insulin Resistance Atherosclerosis Study. J Clin Endocrinol Metab 2015;100:E463–8. https://doi.org/10.1210/JC.2014-2357.

 Guevara-Cruz M, Vargas-Morales JM, Méndez-García AL, López-Barradas AM, Granados-Portillo O, Ordaz-Nava G, et al. Amino acid profiles of young adults differ by sex, body mass index and insulin resistance. Nutr Metab Cardiovasc Dis 2018;28:393–401. https://doi.org/10.1016/j.numecd.2018.01.001.

3. It is important for the authors to clarify why they believe this particular population is the best choice for calculating insulin resistance. Are there any specific risk factors that justify this choice? The choice of age range (18-25 years) should be better justified, considering the higher prevalence of insulin resistance in older individuals.

A. Thank you for your observation. We selected this population based on a previous study where we identified a high prevalence of insulin resistance (IR) within this age range (56.6%) [1]. Our primary focus was on developing a Genetic Risk Score (GRS) to predict the risk of IR, aiming to aid in the prevention of type 2 diabetes in adulthood. We consider it important to evaluate genetic factors by analyzing genetic variants of amino acid metabolism, as we have previously observed a relationship between these variants and IR. Additionally, amino acids could serve as novel biomarkers for IR risk. In situations where insulin levels are elevated but not high enough to manifest as IR, considering other risk biomarkers such as amino acids may contribute to the prediction, prognosis and prevention of these diseases.

Reference

 Guevara-Cruz M, Vargas-Morales JM, Méndez-García AL, López-Barradas AM, Granados-Portillo O, Ordaz-Nava G, et al. Amino acid profiles of young adults differ by sex, body mass index and insulin resistance. Nutr Metab Cardiovasc Dis 2018;28:393–401. https://doi.org/10.1016/j.numecd.2018.01.001.

Methods:

4. The study would benefit from providing more detailed data that would allow other researchers to replicate the study effectively. This would enhance the credibility and scientific value of the findings.

A. Thank you for your observation. We have now provided more details in the Methods section, epecifically, about the genotyping analysis and about the GRS calculation. 

“These 18 SNPs were analyzed using allelic discrimination assays using TaqMan probes (AppliedBiosystems®) by the real time polymerase chain reaction (RT-PCR) on a LightCycler® 480 instrument (Roche®). Briefly, a master mixture was prepared considering for each sample 0.75 µL of TaqMan probe, 0.25 µL of molecular grade nuclease-free ultrapure water (USB®, USA), and 5 µL of Probes Master (LightCycler® 480), following the manufacturer's instructions. Then, to perform PCR, 4 µL of previously adjusted DNA and 6 µL of the master mixture were added to each well of the 96-well plates (Roche®). Negative controls were also included, which only carried the master mixture and nuclease-free water. The reactions were performed in duplicate. The cycling conditions consisted of an initial pre-incubation cycle at 95 °C for 10 min, followed by 45 cycles of denaturation at 95 °C for 12 s, annealing at 60 °C for 50 s and extension at 72 ° C for 2 s and a cooling cycle at 40 °C for 30 s. For allelic discrimination results, the context sequence for each Taqman probe and the fluorophores targeting each allele were previously verified based on information reported by the manufacturer.” 

“This involved multiplying the standardized β coefficient by the effect size (0, 1 or 2) for each SNP, followed by summing the scores obtained from the four SNPs for each subject.

GRS=∑_(i=1)^K▒β_(i ) .N_i

(3)

Where k is the number of independent genetic variants associated with IR, Ni corresponds to the effect size (0, 1 or 2) for each SNP, that is, the number of risk alleles for each individual (i=1), and β is the coefficient estimated for each SNP associated with the HOMA-IR.”

5. The study employed a cross-sectional design with a sample size of 452 subjects aged over 18. Have the authors performed any sample size power calculation?

A. Thank you for your question. We performed the sample size power calculation, which resulted in 89.4%. This information has now been included in the Discussion section.

“Moreover, another limitation of our study lies in its cross-sectional design, which precludes to determine the causality of the results. Further research is required to evaluate whether these SNPs indeed harbor a causal relationship with the development of IR over a time interval. In addition, the study focused on a specific population of young adults, which limits the generalizability of the findings to other age groups or populations. While the power analysis of the utilized sample size exceeded 80%, which is considered acceptable, further research with diverse cohorts would be valuable to validate the observed associations.”

6. The analysis should be adjusted according to sex and BMI, as obesity is related to insulin resistance. 

A. Thank you for your suggestion. We have incorporated this adjustment into our analysis, and the results have been included in the Results section, specifically in Table 4 and Table 5.

Interestingly, subjects with a high GRS showed higher levels of glucose, total cholesterol, triglycerides and insulin levels (p < 0.05) than subjects with a low GRS (cut-off point ≤ 0.624) without covariate adjustment. These results, except for total cholesterol, were maintained when evaluated with adjustment for age, sex and BMI (Table 4).

7. The abstract mentions the use of 18 SNPs for GRS construction, but only 10 were in Hardy-Weinberg equilibrium and included in the models for GRS construction. Finally, only four were used in the association analysis. This should be clearly stated in the abstract to avoid confusion.

A. Thank you for your observation. We have now included this information in the abstract.

“…Eighteen SNPs were genotyped by allelic discrimination. Of these, ten were found to be in Hardy-Weinberg equilibrium, and only four were used to construct the GRS through multiple linear regression modeling.”

Risk allele information:

8. In the supplementary material, it is unclear which allele was referred to as the risk allele in the GRS calculation in the missing information lines.

A. Thank you for bringing this to our attention. We apologize for the omission and have now specified this information in the supplementary material (Supplementary Table 3).

9. In Supplementary Table 1, there is an ambiguous reference for the risk allele. The authors should clarify the intended meaning of the column headings: "PRESUMED REFERENCE RISK ALLELE."

A. Thank you for your observation. We have now modified the column heading in supplementary table 1.

10. How many risk alleles do individuals with higher GRS have? This information is necessary to be useful in clinical practice and replication studies.

A. Thank you for your question. We have now added this information in the Results section.

“The GRS was categorized into tertiles (T1 = 149 subjects; T2= 211 subjects; T3= 92 subjects), revealing that 92 subjects carrying the risk alleles classified in the highest tertile (GRS-high)…”

Discussion:

11. There is a lack of discussion about the association between GRS and amino acid concentration, which is a primary aspect of the manuscript's hypothesis. In my opinion, this manuscript lacks sufficient scientific support to affirm that circulating amino acids are a risk factor for insulin resistance. 

A. Thank you for your thoghtful feedback. We have now included in the discussion that further research is needed to understand how the SNPs used to calculate the GRS impact amino acid concentration. Additionally, we agree with the reviewer that further research is needed to determinate the causal relationship between circulating amino acids and insulin resistance. 

Additional considerations:

12. Provide reference values for blood pressure. 

A. Thank you for your suggestion. We have now included this information in the Methods section. 

13. Use correct abbreviations for cholesterol in lipoprotein particles (e.g., HDL-C, LDL-C).

A. Thank you for your suggestion. We have made the corresponding changes.

14. Include the number of risk alleles individuals with higher GRS have.

A. Thank you very much for your observation. We have now included this information in the results section. 

“The GRS was categorized into tertiles (T1 = 149 subjects; T2= 211 subjects; T3= 92 subjects), revealing that 92 subjects carrying the risk alleles classified in the highest tertile (GRS-high)…”

15. As a limitation, it is important to highlight that the study focused on a specific population of young adults, which may limit the generalizability of the findings to other age groups or populations. Further investigations involving diverse cohorts would be valuable in confirming the observed associations.

A. Thank you very much for your comment. We have added in the study limitations in the discussion section.

“In addition, the study focused on a specific population of young adults, which limits the generalizability of the findings to other age groups or populations. While the power analysis of the utilized sample size exceeded 80%, which is considered acceptable, further research with diverse cohorts would be valuable to validate the observed associations.”

---

## [Decision Letter · Decision Letter 1]

13 Feb 2024

Genetic risk score for insulin resistance based on gene variants associated to amino acid metabolism in young adults

PONE-D-23-27177R1

Dear Dr. Lilia G. Noriega,

We’re pleased to inform you that your manuscript has been judged scientifically suitable for publication and will be formally accepted for publication once it meets all outstanding technical requirements.

Kind regards,

Hongsong Zhang

Academic Editor

PLOS ONE

---

## [Editor Report · Acceptance letter]

21 Feb 2024

PONE-D-23-27177R1 

PLOS ONE

Dear Dr. Noriega, 

I'm pleased to inform you that your manuscript has been deemed suitable for publication in PLOS ONE. Congratulations! Your manuscript is now being handed over to our production team.

Kind regards, 

on behalf of

Dr. Hongsong Zhang 

Academic Editor

PLOS ONE